# Flexible Mechanical Sensors Fabricated with Graphene Oxide-Coated Commercial Silk

**DOI:** 10.3390/nano14121000

**Published:** 2024-06-08

**Authors:** Hyun-Seok Jang, Ki Hoon Lee, Byung Hoon Kim

**Affiliations:** 1Department of Physics, Incheon National University, Incheon 22012, Republic of Korea; slm92jhss@inu.ac.kr; 2Intelligent Sensor Convergence Research Center, Incheon National University, Incheon 22012, Republic of Korea; 3Institute of Basic Science, Incheon National University, Incheon 22012, Republic of Korea

**Keywords:** commercial silk, graphene oxide, thermal treatment, pressure sensor, motion sensor

## Abstract

Many studies on flexible strain and pressure sensors have been reported due to growing interest in wearable devices for healthcare purposes. Here, we present flexible pressure and strain (motion) sensors prepared with only graphene oxide (GO) and commercial silk fabrics and yarns. The pressure sensors were fabricated by simply dipping the silk fabric into GO solution followed by applying a thermal treatment at 400 °C to obtain reduced GO (rGO). The pressure sensors were made from rGO-coated fabrics, which were stacked in three, five, and seven layers. A super-sensitivity of 2.58 × 10^3^ kPa^−1^ at low pressure was observed in the seven-layer pressure sensor. The strain sensors were obtained from rGO-coated twisted silk yarns whose gauge factor was 0.307. Although this value is small or comparable to the values for other sensors, it is appropriate for motion sensing. The results of this study show a cost-effective and simple method for the fabrication of pressure and motion sensors with commercial silk and GO.

## 1. Introduction

Flexible electronic components are steadily being developed for various applications, including electrically conductive textiles [1,2,3,4], thermoelectric textiles [5,6,7], solar cells [8], secondary batteries [9], and gas sensors [10,11,12]. Among them, flexible mechanical sensors (strain and pressure sensors) have been of interest because of their utility for sensing human body functions for healthcare purposes and for use in applications related to the Internet of Things (IoT). In particular, textile-based mechanical sensors have the advantage of being lightweight and low-cost and having superior flexibility [13,14,15,16]. Commercial textiles such as cotton, nylon, polyester, and silk have been used as flexible substrates for such sensors. However, other components are used to ensure high sensitivity and flexibility.

Cotton fabric has been decorated with Ag nanowires, Ag flowers, and Ag thread [17,18,19,20]; coatings of electrochemically exfoliated graphene film have been applied on cotton fabric [21]; and coatings of Ag have been applied to cotton yarn using Ag paste followed by sintering [22] to provide electrical conductivity. Likewise, electronic textiles for sensors fabricated with nylon and polyester have been created using conductive polymers [23], Au and MoS_2_ coating [24], graphite-polyurethane (PU) coating applied to crochet knitted elastic [25], the screen printing of elastic conductive carbon ink on polyester fabric [26], the wet spinning of CNT/thermoplastic PU on silver-plated nylon electrodes [27], the coating of CNT/carbon black PU on fabric with Ag-coated conductive nylon fiber [28], and the spraying of multiwall CNT/TiO_2_ conductive suspensions on nylon textiles [29]. Mxene has also been applied to textiles to improve sensitivity [30,31,32,33,34]. In the previous studies mentioned above, complex processes are inevitably required for the fabrication of the sensors.

In the case of silk, C. Wang et al. reported that wearable strain sensors fabricated using carbonized silk fabric had a large gauge factor, but an elastic polymer was used due to the thermal instability of carbonized silk [35]. A pressure sensor was fabricated using Ag nanowires coiled onto silk-fiber-wrapped PU [36], 3D graphene oxide (GO), and a silk fibroin mixture thermally reduced at 700 °C [37]. However, its sensitivity was very low (0.136~0.54 kPa^−1^).

Graphene has attracted attention due to its unique properties and great potential for use in diverse fields [38,39,40,41]. However, the need to obtain large quantities of graphene has hindered it from being adopted for various applications. The use of GO is one of the most promising methods of solving this problem. Mechanical sensors fabricated with GO have been reported [37]. Z. Yang et al. reported the production of direction-dependent strain sensors using GO and polyester [42]. Twisted graphene fiber for motion sensing was created using GO and elastic Lycra warp yarns [43]. Strain sensors created using rGO and ecoflex showed low sensitivity (S = 0.122 kPa^−1^) and high GF (31.6 at 400% strain) [44]. Using GO-coated PU and carbonization followed by polypyrrole attachment, R. Wang et al. synthesized a pressure sensor whose GF was 0.770 near 40% strain [45]. Even though the mechanical sensors were devised successfully, their fabrication also involved a complicated procedure and needed multiple ingredients.

Here, we report flexible pressure and strain sensors fabricated using reduced GO (rGO)-coated commercial silk (rGOS) fabrics and yarns. The GO was coated using a simple dipping method, and then the GO-coated silk (GOS) was thermally reduced at 400 °C. The pressure and strain sensors were prepared by stacking small pieces of rGOS fabrics and twisting rGOS yarns, respectively. The pressure sensor showed superior sensitivity (2.58 × 10^3^ kPa^−1^ at low pressure and 1.842 kPa^−1^ at high pressure). The gauge factor (GF) of the strain sensor was relatively low (~0.307), but the strain sensor was able to detect human motion (bending) and had good cyclability even after 5000 cycles.

## 2. Materials and Methods

GO was synthesized with graphite powder (99.995% metals basis, Alfa Aesar, Ward Hill, MA, USA) using the modified Hummers method [46]. GO was dissolved in DI water and sonicated for 1.0 h. The GO concentration was 1.0 mg/mL. GO was coated onto commercial silk fabrics by simply dipping them for 30 min, and then the GOS fabric was dried at 40 °C under a fume hood. This process was repeated three times to produce a uniform coating of GO [2,12]. The GOS fabric was thermally reduced. The rGOS fabrics for the pressure sensors were obtained via thermal treatment at 400 °C for 2.0 h at a heating rate of 1.0 °C/min in N_2_ atmosphere. The rGOS fabrics were cut into 1.0 cm × 1.0 cm pieces, and then three pressure sensors were made depending on the number of fabrics. The strain sensors were fabricated by twisting the GOS yarns with an electrically powered drill. After twisting the GOS yarns, they were thermally treated under the same conditions.

To observe the morphology of the samples, field-emission scanning electron microscopy (FE-SEM, JSM-7800F, JEOL, Akishima, Tokyo, Japan) was used. The structure of the rGOS was determined using X-ray diffraction (XRD, SmartLab, Rigaku, Tokyo, Japan) with Cu Kα radiation (λ = 1.5418 Å), X-ray photoelectron spectroscopy (XPS, PHI 5000 Versa Probe II, Ulvac-PHI 5000 Versa Probe, Phi(Ø), Chigasaki, Japan), and Raman spectroscopy (Raman-LTPL system, Witec alpha300, Witec, Ulm, Germany) using a 532 nm laser. The size of the GO was measured using atomic force microscopy (AFM, XE-NSOM, Park Systems, Suwon, Korea). Sensing performance was measured using a Keithley 6221 current source, Keithley 6517 digital multimeter, and Keithley 2182A nanovoltmeter (Keithley, Solon, OH, USA).

## 3. Results and Discussion

### 3.1. Structural Investigation

The as-synthesized GO samples were dispersed in DI water and then dropped onto a SiO_2_/Si wafer to measure the size of the GO using AFM (Figure 1a). The distribution of the lateral size was from 50 nm to 1.95 μm (Figure 1b). The average size of the GO samples was 349 ± 152 nm (the average particle size and its standard deviation are shown in Appendix A). Figure 1c,d show optical images of the GOS and rGOS, respectively. After thermal treatment, the dark brown color of the GOS changed to black (rGOS), and the size of the fabric was reduced. The shrinkage of the fabric was also observed in SEM images of the GOS (Figure 1e) and rGOS (Figure 1f). Wrinkles on the surface of the silk were observed, demonstrating that GO had been coated on the silk surface (the inset of Figure 1f).

The Raman spectroscopy examination of GOS and rGOS confirmed the existence of GO on the silk (as shown in the inset of Figure 1e). *D* and *G* peaks were observed in the GO, GOS, and rGOS. The *D* band arises from defects (disorder), and the *G* band is related to C-C stretching of the *sp^2^* carbon [47]. The *D* peaks were at 1341, 1359, and 1330 cm^−1^ and the *G* peaks were at 1584, 1587, and 1558 cm^−1^ for the GO, GOS, and rGOS, respectively. The peak shifts in the GOS compared with GO are attributed to overlapping with the peaks from the silk. After thermal treatment (rGOS), the *D* and *G* peaks were red-shifted, moving from 1359 cm^−1^ to 1330 cm^−1^ for the *D* peak and from 1587 cm^−1^ to 1558 cm^−1^ for the *G* peak. This indicates the increase in the translational symmetry of the GOS due to the thermal treatment. This behavior was also observed for the *I_D_*/*I_G_* ratio. The *I_D_*/*I_G_* ratios of the GO, GOS, and rGOS were 1.06, 1.06, and 1.01, respectively.

Figure 2a displays the normalized XRD patterns of the GO sheet, GOS, and rGOS. The peak at 11.44° ((001) plane) of the GO sheet indicates that the interlayer distance of the GO is 7.732 Å. The XRD patterns of the GOS were similar to those of pristine silk [48]. After heat treatment, the peak for the (002) lattice plane at 22.95° appeared in the rGOS. The relatively broad peak for the (002) plane originated from the overlapping of various amorphous structures [49].

The decrease in the number of oxygen functional groups due to thermal treatment was investigated using the XPS C1s peak (Figure 2b–d). A large number of oxygen functional groups (COOH, C=O/O−C=O, and C−O/C−O−C/C−OH) were clearly observed in the GO sheet (Figure 2b) and GOS (Figure 2c). In contrast, the number of functional groups significantly decreased in rGOS (Figure 2d). At the O1s peak, the quantities of O−H, C−O, and C=O species in the GOS were similar to those in the GO sheet, but the number of C−O bonds decreased in the rGOC (Appendix A). The nitrogen from the silk was found in the N1s peaks of the GOS and rGOS (Appendix A). Interestingly, pyridinic N developed after thermal treatment, resulting from the carbonized silk. The XRD, Raman spectroscopy, and XPS studies showed the reduction of GO and the carbonization of silk, which endow electrical conductivity to rGOS.

### 3.2. Pressure Sensors

Figure 3a is a side view of a single piece of rGOS fabric. This fabric’s average thickness is 349.6 μm. Flexible pressure sensors were made very simply using the rGOS fabrics. First, the fabric was cut into a square, whose length on one side was 1.0 cm. Next, the cut fabrics were stacked between Al plates (Figure 3f). We chose to use three, five, and seven layers of fabric. The thickness of the seven-layer pressure sensor was about 4.52 mm (Figure 3b).

The sensitivity of the three sensors is depicted in Figure 3c–e. The sensitivity of the three-layer pressure sensor was 3.32 kPa^−1^ at low pressures (0.5–21 kPa), 0.662 kPa^−1^ at 21–46 kPa, and 0.665 × 10^−3^ kPa^−1^ at high pressures (46–290 kPa) (Figure 3c). For the five-layer pressure sensor, the sensitivity was 18.7 kPa^−1^ at low pressures (0.4–10.9 kPa), 3.82 kPa^−1^ at 10.9–91.3 kPa, and 2.57 × 10^−3^ kPa^−1^ at high pressures (91.3–402.1 kPa) (Figure 3d). Interestingly, an enormous increase in sensitivity was measured in the seven−layer pressure sensor. The sensitivities were 2.58 × 10^3^ kPa^−1^ at low pressures (0.2–72.3 kPa) and 1.84 kPa^−1^ at high pressure (72.3–421 kPa) (Figure 3e). To the best of our knowledge, the sensitivity at low pressure of the seven-layer sensor is the best among all textile-based sensors, except for that of a pressure sensor fabricated with Ag nanowire (AgNW)−coated cotton fabric [17] (Table 1). The operation of the sensor is shown in Figure 3f. Without pressure, the resistance (*R*) of the sensor was large because the rGOS layers barely contacted each other, i.e., the charge carriers were isolated in each layer due to a large potential barrier in the vertical direction (as shown in the left panel in Figure 3f). As soon as pressure was applied, the layers contacted each other. This led to small potential barriers between the layers. Hence, the carriers freely moved between the layers, resulting in a significant decrease in the *R* (as shown in the right panel in Figure 3f). The amount of layer-dependent sensitivity was determined by the initial *R* of the sensors. The increase in layer number increased the initial *R*. For example, the resistances of the three-layer and seven-layer sensors were 0.42 MΩ and 958 MΩ, respectively.

Drawing on these results, we fabricated a two-by-two pressure module using the rGOS fabrics (0.5 × 0.5 cm^2^) and Au wires. The surface of the module was protected with pristine silk fabric (Figure 3g,h). The module directly measured the applied force, indicated by different colors using the Labview program (Appendix A). Figure 4 displays measurements of the pressures of the four sensors in the module. The numbers in the red and white boxes indicate the applied forces. The response time of the sensors was 300 ms (Appendix A).

To check the cyclability of the pressure sensors, 2.0 kPa of pressure was applied to the seven-layer sensor using a homemade pressure machine with a step motor. When the pressure was applied, the current increased to near 50 μA. This value was maintained over 1000 cycles (Figure 5a). Figure 5b,c display variations in current near the first and 1000 cycles. It shows that the pressure sensor has good repeatability.

### 3.3. Motion Sensors

The strain sensors were fabricated with GOS yarns. First, the GO-coated yarn was twisted using an electrically powered drill (Figure 6a). Figure 6b shows the variation in force while twisting the yarn. Before being twisted, the force was unstable, but the force was relatively stable after twisting began (yellow box). Then, the twisted GOS yarn was thermally treated using the same conditions as those induced for fabricating the rGOS fabric. Evenly twisted yarn can be observed in the SEM image (Figure 6c).

Figure 7 depicts the strain-dependent *R*. As the twisted yarn was stretched from 4.40 cm (*L*_0_) to 7.15 cm (162.5%) (Figure 7a), the *R* changed from 2.295 MΩ (*R*_0_) to 1.855 MΩ (Figure 7b). This decrease in *R* can be explained as follows. When the yarn was stretched, the diameter of the twisted yarn decreased from 543.2 μm at 0% strain to 377.9 μm at 160% strain (Appendix A). This means that the rGOS fibers in the stretched yarn contacted each other much better compared to the degree of contact for the unstretched yarn, causing an increase in electrically conductive paths (Figure 7d). Although the *R* slightly increased (*R* = 2.308 MΩ, see the red triangle in Figure 7), *R* almost recovered when the strain sensor was released again. The increase in *R* was caused by the breaking of some parts of the fibers when the yarn was stretched to its maximum, as shown in Appendix A. The variation in *R* was 19.17% at 162.5% stretching. The GF can be derived as follows: (Δ*RL_0_*)/(*R_0_*Δ*L*). The GF of the strain sensor was small (GF = 0.307) compared with that reported in previous studies [25,29,33,35], but it was larger than that of cotton yarn produced using Ag-paste [22] and Ag-coated nylon [50]. Hence, we found that the twisted rGOC yarn could be used as a motion sensor.

Figure 8 shows the detection of finger motion. The twisted rGOS yarn was attached to a Latex glove with Ag foil to monitor the motion of a finger (Figure 8a). The current was approximately 240 nA when the subject’s fingers were spread out. When one finger was bent about 90°, the current increased to ~280 nA. The change in the current was about 16.7% (Figure 8b). This indicates that the twisted rGOS yarns functioned as a motion sensor. To evaluate the cyclability for bending, we made a hand using 3D printing with a motor (Figure 8c and Appendix A). The variation in current (in the ratio Δ*I*/*I*_0_, *I*_0_ is the current corresponding to stretching, and Δ*I = I* − *I*_0_) for bending and stretching was stable for 5000 cycles (Figure 8d). Figure 8e,f display variations in current near 2500 cycles and 5000 cycles, respectively, indicating the good cyclability of the motion sensor even after 5000 cycles. The difference in current variation between the human finger (16.7%) and the artificial finger (~3.5%) was due to the bending degree. The human finger was bent 90°, but the artificial hand could only be bent to a maximum degree of about 45°. The double peaks during bending were caused by two finger joints, as shown in Figure 8c (white circles).

## 4. Conclusions

In summary, we fabricated flexible pressure and motion sensors using GO and commercial silk fabric and yarns, respectively. The sensors were simply produced by dipping the silk into the GO solution followed by thermal treatment. The pressure sensors were created by stacking the rGOS fabrics. The sensitivity of the seven-layer pressure sensor was 2578 kPa^−1^. As shown in Table 1, this value is very high compared to the values exhibited by other flexible pressure sensors. The GF of the motion sensors obtained from the twisted rGOS yarns was 0.307. The motion sensors successfully detected the bending of a finger, and the response for bending and stretching was stable for up to 5000 cycles. This study provides a method of synthesizing ultrasensitive pressure and motion sensors without complicated procedures.

## Figures and Tables

**Figure 1 nanomaterials-14-01000-f001:**
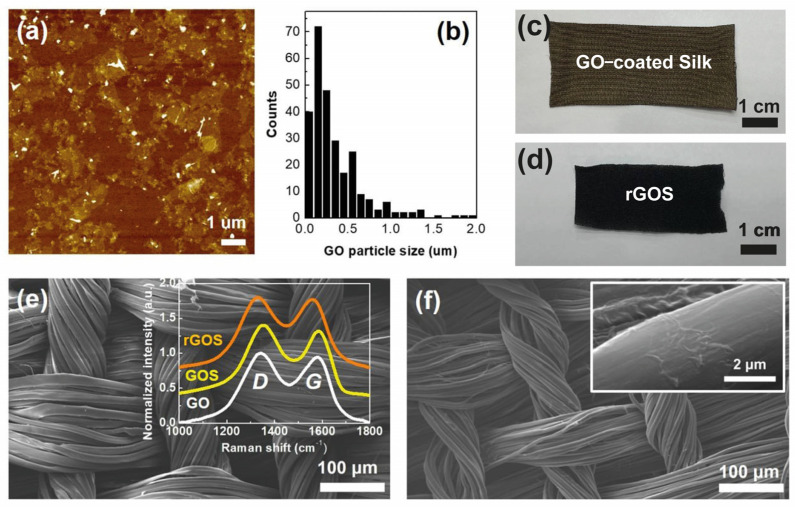
(**a**) AFM image and (**b**) the sizes of the GO particles. Optical images of (**c**) GOS and (**d**) rGOS fabrics. SEM images of (**e**) GOS and (**f**) rGOS fabrics. The inset in (**e**) shows the Raman D and G peaks of GO, GOS, and rGOS. The inset in (**f**) shows that GO samples were well coated onto the surface of the silk.

**Figure 2 nanomaterials-14-01000-f002:**
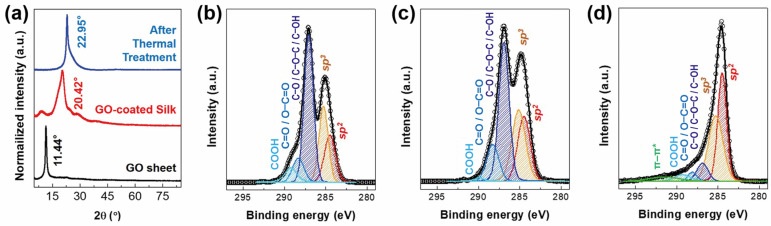
(**a**) XRD patterns of the GO, GOS, and rGOS. XPS C1s peaks of (**b**) GO, (**c**) GOS, and (**d**) rGOS. A decrease in the number of oxygen functional groups was observed in the rGOS.

**Figure 3 nanomaterials-14-01000-f003:**
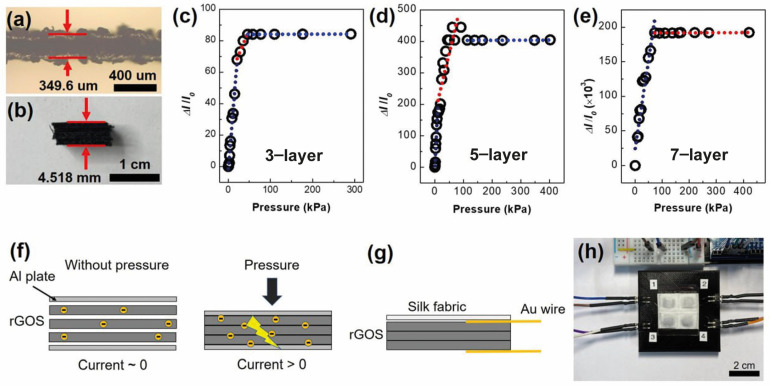
Side−views of (**a**) a single rGOS fabric and (**b**) a 7−layer rGOS fabric. The sensitivities of the rGOS pressure sensors fabricated with (**c**) 3−layer, (**d**) 5−layer, and (**e**) 7−layer fabrics. (**f**) The operating mechanism of the pressure sensor. The yellow circles are charge carriers. (**g**) A schematic illustration of the textile−based pressure sensor module and (**h**) an optical image of the two−by−two pressure sensor module.

**Figure 4 nanomaterials-14-01000-f004:**
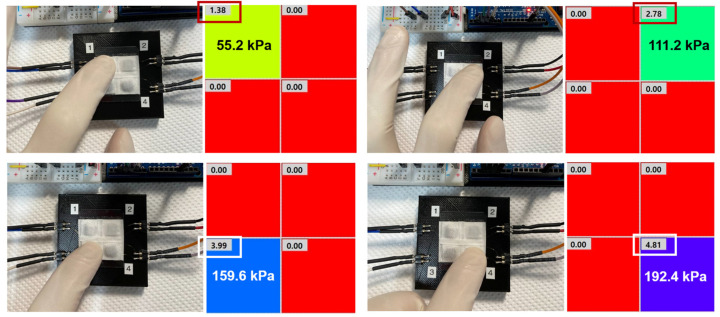
Real-time pressure sensing of each sensor in the module depicted using the Labview program.

**Figure 5 nanomaterials-14-01000-f005:**
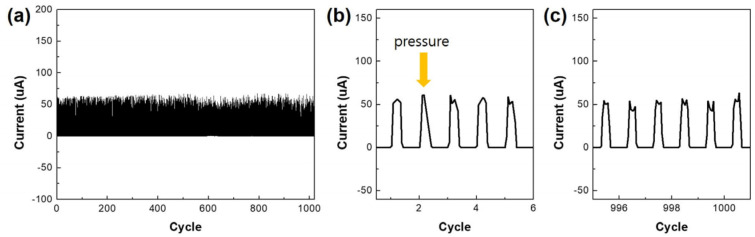
(**a**) Repeatability of the 7−layer pressure sensor during 1000 cycles and near (**b**) the first cycles and (**c**) 1000 cycles.

**Figure 6 nanomaterials-14-01000-f006:**
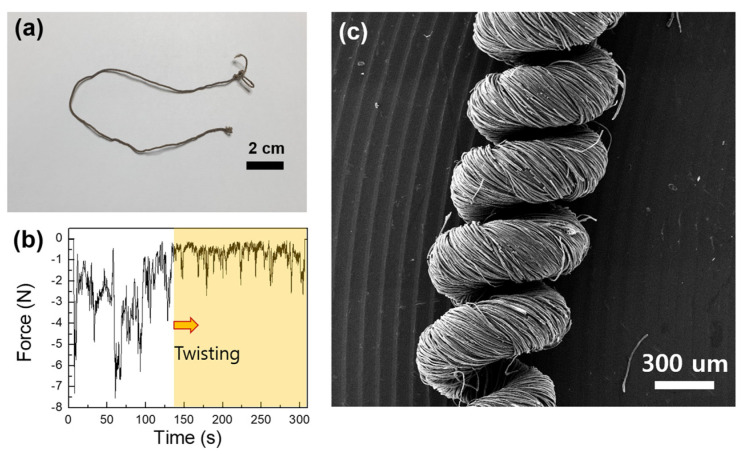
(**a**) Optical image of the twisted rGOS yarn. (**b**) The variation in force during the twisting of the yarn. (**c**) SEM image of the twisted rGOS yarn.

**Figure 7 nanomaterials-14-01000-f007:**
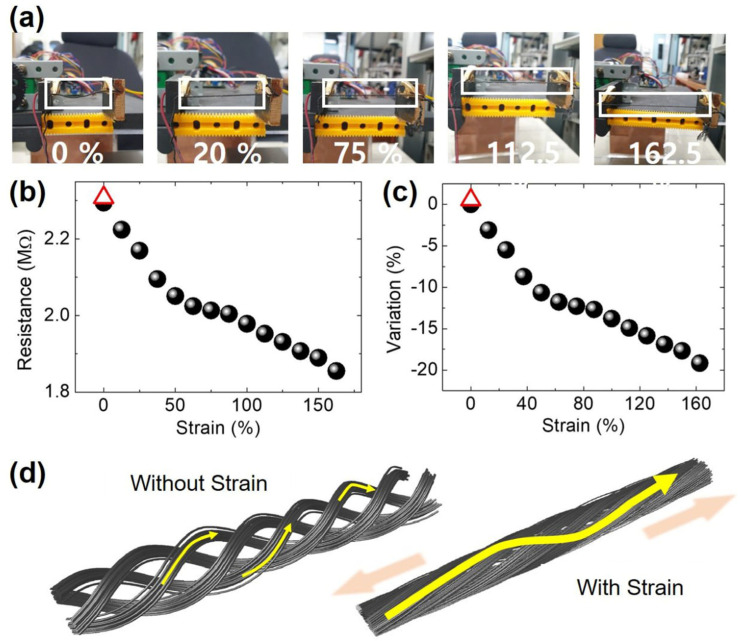
(**a**) Optical images of the application of strain to the twisted rGOS yarn (white boxes). The yarn was maximally stretched by 162.5%. (**b**) The strain-dependent *R* and (**c**) Δ*R*/*R*_0_ (variation); here, Δ*R* = *R* − *R*_0_, and R_0_ is the initial R (when the strain is zero). The negative value means that the *R* decreased as the strain increased. (**d**) Schematic illustration of the decrease in *R* due to strain. Yellow arrows are the electrically conductive paths.

**Figure 8 nanomaterials-14-01000-f008:**
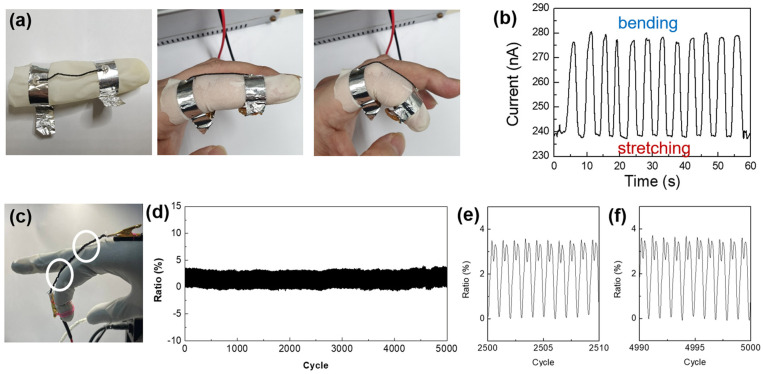
(**a**) Test of the motion sensor using a human finger and (**b**) its result. The current increases as the finger bends. (**c**) An artificial hand fabricated using a 3D printer. (**d**) Cyclability of the yarn during 5000 cycles and near (**e**) 2500 cycles and (**f**) 5000 cycles.

**Table 1 nanomaterials-14-01000-t001:** Comparison of characteristics of the pressure sensors.

Textile	Materials/Methods	Sensitivity (kPa^−1^)	Ref.
Silk	Silkworm cocoon, GO, C_6_H_12_O_6_, NaOH/hydrothermal and freeze-drying	0.54~0.21	[37]
Silk	Silkworm fiber, AgNW, PU, CNT/coiling of AgNW onto silk-fiber-wrapped PU yarn	0.136	[36]
Nylon, cotton, silk	Nylon, cotton, silk, PU filament/application of Ag coating on nylon yarn using ion-plating method, wrapping of cotton yarn around PU filament and silk around Ag-coated nylon yarn using covering machine	0.01884	[50]
Cotton	Cotton fabric, AgNW, FeCl_3_, ethylene glycol, poly(vinylpyrrolidone), Ag paste/synthesis of AgNW using a hydrothermal method, fabrication of Ag paste on cotton fabric using a screen-printing process, coating of AgNW on cotton via a soaking process	4.42 × 10^3^, (Comparable to the results in this study)	[18]
Cotton	Cotton fabric, AgNW, isopropyl alcohol, polyvinyl pyrrolidone, ethylene glycol, AgNO_3_, NaCl/synthesis of AgNW using a hydrothermal method, application of AgNW coating on cotton via dipping	2.16 × 10^4^	[17]
Cotton	Knitted cotton/spandex fabric (KSCF), pyrrole, ferric chloride hexahydrate, sodium 5-sulfosalicylate, cyclohexane, AgNO_3_, sodium citrate, KNO_3_, Na_2_SO_4_/synthesis of PPy@KSCF using in situ polymerization method, preparation of PPy@KSCF via Ag flower using an electrodeposition process	17.41	[19]
Cotton	Cotton, graphite, stainless steel, PDMS, (NH_4_)_2_SO_4_, NMP, NaOH/ electrochemical exfoliation of graphene (EEG), coating of EEG films on cotton using a hot press	0.16	[21]
Cotton	Cotton fabric, polyester, Kapton tape, sodium dodecylbenzene sulfonate, VHB film, CNT, application of Ni/CNT coating on cotton fabric via dipping, patterning using CO_2_ laser scribing, Ni-coating with the aid of a mask via electroless plating	14.4	[51]
Polyester	Polyester, MoS_2_, HAuCl_4_, ethanol/treatment of the fiber with air plasma, applying a MoS_2_ coating via dropping, immersion of the fiber in an aqueous HAuCl_4_ to grow Au nanostructure	0.19	[24]
Polyester	Polyester fabric, carbon ink, waterborne PU (WPU), CNT, Venetian fabric, CNT paste/coating of carbon ink on polyester via screen printing and curing at 100 °C, mixing cationic WPU and anionic WPU with a CNT suspension, electrostatic deposition of WPU/CNT via dipping of Venetian fabric into WPU/CNT suspensions	3.42	[26]
Silk	GO, application of commercial silk/GO coating on silk using a dipping method, applying a 400 °C thermal treatment to reduce GO	2.58 × 10^3^	This work

## Data Availability

Data are contained within the article.

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
