# Peer review of "Flexible Mechanical Sensors Fabricated with Graphene Oxide-Coated Commercial Silk"

_nanomaterials, 2024, doi:10.3390/nano14121000_

Round 1
Reviewer 1 Report
Comments and Suggestions for Authors
In this paper, the authors report a flexible pressure and strain (motion) sensor prepared using graphene oxide (GO) and commercial silk fabrics and yarns. The results of this study show a cost-effective and simple method for the fabrication of pressure and motion sensors with commercial silk and GO. I believe that publication of the manuscript may be considered only after the following issues have been resolved.
1. In the abstract section, the relevant background introduction should be as few as possible, after all, the introduction section has already been elaborated in detail.
2. The author mentioned commercial silk coated with graphene oxide in the paper, but there is no evidence to suggest that it is a coated structure.
3. What is the physical mechanism behind the superior performance of flexible mechanical sensors prepared from commercial silk coated with graphene oxide? Suggest the author to use relevant diagrams to illustrate
4. There are still many reports on the use of graphene oxide for flexible mechanical sensors, but the author did not provide a detailed discussion in the introduction section.
5. In order to increase the readability of the article, in the introduction section, the author needs to mention some relevant latest references on the application of graphene-based material, such as, Diamond and Related Materials 142, 2024, 110793; Opto-Electron Sci 2, 230012 (2023).
6. The English expression of the whole article needs to be further improved.
Comments on the Quality of English LanguageMinor editing of English language required
Author Response
Dear Reviewer 1.
We would like to present our acknowledgment of your instructive comments. After deep consideration, we made a response as the attached file.

Reviewer 2 Report
Comments and Suggestions for Authors
The authors demonstrated flexible pressure and strain sensors with good sensitivity prepared with graphene oxide and commercial silk fabrics and yarns. The results are interesting and there are some issues need to be addressed.
1. What is the thickness of the rGO coated on the fabrics?
2. What about the repeatability and the consistency of the rGO coated fabric pressure sensor? 3.Why did the twisted yarn showed a trend of decreasing resistivity during tensile loading to ~160%,and the deformation process of the twisted yarn should be studied with in situ microscope methods to fully illustrate the mechanism. 4.The figure caption for Figure 3(h) should be corrected.Author Response
Dear Reviewer 2,
We would like to present our acknowledgment of your instructive comments. After deep consideration, we made a response as the attached file.

Reviewer 3 Report
Comments and Suggestions for Authors
The manuscript titled "Flexible Mechanical Sensors Fabricated with Graphene Oxide-Coated Commercial Silk" presents an innovative and straightforward method for fabricating flexible pressure and strain sensors using reduced graphene oxide (rGO) and commercial silk. The study demonstrates the high sensitivity and good cyclability of these sensors, which have significant potential for applications in wearable devices and the Internet of Things (IoT). The experimental results are detailed, and the discussion is thorough. While the manuscript is well-prepared and the findings are significant, there are a few areas where additional data and analysis could further enhance the study:
1. For AFM imaging, it is suggested to increase the statistical sample size and provide the standard deviation of particle size distribution. This will make the size distribution data more representative and statistically significant. Conducting AFM imaging on a larger number of GO particles and calculating the average particle size along with its standard deviation will provide more robust data.
2. In SEM measurements, it is recommended to use Energy Dispersive X-ray Spectroscopy (EDS) to analyze the elemental composition in SEM images. EDS analysis can confirm the uniform distribution of GO on silk, providing more detailed information about the elemental composition and ensuring the consistency of the coating process. Performing EDS analysis during SEM imaging to map the distribution of elements such as carbon and oxygen will confirm the uniform coating of GO on the silk fibers.
In summary, this manuscript provides a valuable contribution to the field of flexible sensor fabrication using GO-coated silk. The suggested improvements regarding AFM statistical analysis and EDS in SEM measurements will enhance the robustness and comprehensiveness of the study. I look forward to seeing these additions in the revised manuscript.
Author Response
Dear Reviewer 3,
We would like to present our acknowledgment of your instructive comments. After deep consideration, we made a response as the attached file.

Round 2
Reviewer 1 Report
Comments and Suggestions for Authors
Accept in present form
Reviewer 3 Report
Comments and Suggestions for Authors
Thank you for your response to my review comments. I have read your reply and I agree with your revisions. Your proactive attitude and careful amendments significantly contribute to enhancing the quality of the manuscript. I look forward to seeing the final version of the manuscript and hope it will be successfully published.